# Self-Supervised Emotion Representation Disentanglement for Speech-Preserving Facial Expression Manipulation

## ABSTRACT

Speech-preserving Facial Expression Manipulation (SPFEM) aims to alter facial emotions in video content while preserving the facial movements associated with speech. Current works often fall short due to the inadequate representation of emotion as well as the absence of time-aligned paired data—two corresponding frames from the same speaker that showcase the same speech content but differ in emotional expression. In this work, we introduce a novel framework, Self-Supervised Emotion Representation Disentanglement (SSERD), to disentangle emotion representation for accurate emotion transfer while implementing a paired data construction module to facilitate automated, photorealistic facial animations. Specifically, We developed a module for learning emotion latent codes using StyleGAN's latent space, employing a cross-attention mechanism to extract and predict emotion editing codes, with contrastive learning to differentiate emotions. To overcome the lack of strictly paired data in the SPFEM task, we exploit pretrained StyleGAN to generate paired data, focusing on expression vectors unrelated to mouth shape. Additionally, we employed a hybrid training strategy using both synthetic paired and real unpaired data to enhance the realism of SPFEM model's generated images. Extensive experiments conducted on benchmark datasets, including MEAD and RAVDESS, have validated the effectiveness of our framework, demonstrating its superior capability in generating photorealistic and expressive facial animations.

## CCS CONCEPTS

• **Computing methodologies → Image manipulation**.

## KEYWORDS

Emotion Representation Disentanglement, Self-Supervision, Expression Manipulation, Lip Synchronization

## 1 INTRODUCTION

Speech-preserving facial expression manipulation (SPFEM) aims to adjust emotional expressions while maintaining the natural movement of the mouth in videos. This capability significantly enhances human expressiveness, offering substantial benefits to a variety of applications such as virtual avatars and film/television production. Traditionally, capturing the precise emotional expressions of actors requires considerable effort, including numerous

*ACM MM, 2024, Melbourne, Australia*
© 2024 Copyright held by the owner/author(s). Publication rights licensed to ACM.
ACM ISBN 978-x-xxxx-xxxx-x/YY/MM
https://doi.org/10.1145/nnnnnnn.nnnnnnn

takes and extensive post-production work. However, with a robust SPFEM system, it is possible to effortlessly modify facial emotions to achieve the desired effect during post-production. This innovation not only streamlines the filmmaking process but also opens up new possibilities for creative expression, making it a highly anticipated advancement in the industry.

Previous studies have primarily focused on adapting existing face reenactment algorithms [31] or merging the identity and expressions from target and source videos respectively [22] to tackle this challenge. However, these approaches fall short due to the absence of paired data, i.e., recordings of the same script by the same actor with varying expressions. Even with the recording of such data, there's no guarantee that video frames corresponding to different emotions will align on a one-to-one basis, leading to inaccuracies in supervision. Consequently, it leads to sub-optimal results in the following two aspects. First, current approaches are hindered by an inadequate representation of emotion information, limiting their ability to accurately manipulate facial expressions based on reference images. Second, they cannot well preserve the facial animation of the original speech due to the lack of mouth shape guidance aligned with reference emotions.

To address the issues discussed above, we propose focusing on two primary aspects. Firstly, disentangling emotion representation is crucial. This entails the efficient extraction of emotional cues from a reference face and their precise transfer to the source face's emotional representation. In this way, we ensure the provision of a solid and accurate informational foundation for the creation of face images that aligned with reference emotions. Secondly, developing an accurate paired supervision mechanism is essential. This mechanism aims to automatically steer the model towards accurately generating faces based on emotional representations without additional manual annotation. With meticulous frame-by-frame supervision, we ensure that the generated face images not only visually embody specific emotions but also exhibit motion coherence with the source speech content, leading to more realistic and expressive facial animations.

To this end, we propose a novel Self-Supervised Emotion Representation Disentanglement (SSERD) framework, which learns decoupled emotion representation and establishes paired supervision in a self-supervised manner. Specifically, we first design a contrastive emotion latent code learning module, which harnesses the latent space of StyleGAN [17] for efficient representation of emotion information. This module employs a cross-attention mechanism to extract emotion information from reference latent codes and to predict editing codes as residuals relative to the original latent codes. To ensure capturing emotional information, we introduce contrastive learning that promotes proximity between editing codes linked to identical emotions, while simultaneously discouraging the closeness of codes associated with divergent emotions. Next, we propose a paired data construction module

to address the challenge posed by the absence of strictly paired data in the SPFEM task. This module leverages a pretrained StyleGAN to generate accurately paired data by identifying and modifying expression editing vectors that are independent of mouth shape. Moreover, to narrow the gap between synthetic data and real data, we adopt a hybrid training strategy that utilizes both type of data for training. This training strategy aims to improve the realism of the images generated by the SPFEM model.

In summary, the contributions of this work can be summarized as follows: 1) A novel Self-Supervised Emotion Representation Disentanglement (SSERD) framework is proposed. This framework is capable of producing photorealistic videos that feature natural expressions and lip movements. 2) A contrastive emotion latent code learning module is designed for efficient representation of emotion information within the latent space of StyleGAN[17]. 3) A paired data construction module is developed to automatically synthesis paired data via a pretrained StyleGAN, which facilitates the training of SPFEM task. 4) Extensive experiments have been conducted on various benchmarks (e.g. MEAD [34], RAVDESS[20]), demonstrating the superior performance of the proposed framework. *We will release the codes and trained models to facilitate SPFEM research.*

## 2 RELATED WORKS

In this section, we review three key areas related to the task of SPFEM: video-based face manipulation, facial reenactment, and speech-preserving facial expression manipulation.

**Video-based face manipulation.** Video-based face manipulation methods [12, 18, 21, 24] usually employ conditional Generative Adversarial Networks (GANs) [11] or 3D facial models (for example, 3D Morphable Models (3DMM) [2]) to adjust the attributes of speaking faces. It facilitates the conversion of visuals across varied realms, preserving the essence of the original footage. For example, GANimation [24] introduces a GAN conditioning approach using Action Units [9] annotations, effectively mapping the facial movements that characterize human expressions on a continuous spectrum. Tzaban et al. [32] leverages StyleGAN [15] alignment and the neural networks' affinity for learning low-frequency functions to solve temporal consistency issues. Liu et al. [29] manipulate facial expressions in videos by separately representing and estimating the 3D facial structure and movement. Gan et al. [10] employ a pretrained emotion-agnostic talking head transformer and incorporate adaptation modules to facilitate accurate emotional manipulation. SPFEM poses a greater challenge than traditional face alteration methods as it demands the alteration of facial expressions without compromising the original speech animations.

**Face reenactment.** This involves mimicking the speech and facial expressions from a source video [19, 26, 31, 35, 37]. 2D-based techniques produce the desired image directly. For instance, ICface [31] manipulates the pose and facial expressions based on interpretable control signals like head pose angles and action unit values. FOMM [27] separates appearance and motion using a set of key points and their affine transformations. StyleHEAT [39] employs a video motion generation module and an audio-based one to alter StyleGAN's latent features for visual animation, while

a calibration network addresses transformation distortions. Some approaches use 3DMM for separating expression and identity, using a neural renderer for 2D image mapping. Head2Head++ [7] captures complex facial movements and synthesizes temporally cohesive videos with a sequential generator and a specialized dynamics discriminator. SPFEM extends beyond expression manipulation by also needing to maintain the mouth movements of the original speech, adding to its complexity.

**Speech preserving facial expression manipulation** SPFEM's goal is to alter a source video to reflect a target emotion, all while maintaining the speech-related facial movements. Previous approaches, adapted facial reenactment techniques for SPFEM such as ICface [31]. However, these could alter expressions and mouth shapes together, failing to preserve speech accurately. More recently, Papantoniou et al. [22] suggested blending the 3DMM parameters of the source identity and target emotion for this purpose. Sun et al. [30] use 3DMM to capture emotional facial movements and use StyleGAN to model texture map to capture appearance details. Despite significant progress have been made by these works, achieving photorealistic quality and natural expressions remains a challenge due to inadequate emotion representation and the lack of time-aligned paired data. Our work leverages StyleGAN's latent space for comprehensive emotion representation and its editable nature to create high-quality paired training data, addressing this issue effectively.

## 3 METHODOLOGY

The Self-Supervised Emotion Representation Disentanglement (SSERD) framework learns decoupled emotion representation and establishes paired supervision via a pretrained StyleGAN. SSERD contains three key compoments: First, the contrastive emotion latent code learning (CELCL) module leverages the cross-attention mechanism to extract emotion information from reference images. Second, the paired data construction module harnesses StyleGAN's robust generative capabilities and the editable nature of its latent space to generate paired data. Third, the hybrid training strategy utilizes both synthetic paired data and real unpaired data for training to narrow the gap between synthetic images and real images. An overall illustration is presented in Figure 1.

### 3.1 Contrastive Emotional Latent Code Learning

The CELCL module is designed to disentangle emotion information from reference images and predicts emotion latent codes as residual to the source latent codes. It leverages StyleGAN's [17] latent space for comprehensive emotion representation. Concretely, given a source image $I^s$ and a reference image $I^r$, we first utilize a latent encoder $E$, implemented by IResNet50 [8], to derive source latent code $\omega^s \in \mathbb{R}^{18 \times 512}$ and reference latent code $\omega^r \in \mathbb{R}^{18 \times 512}$, formulated as

$$\omega^s = E(I^s)$$
$$\omega^r = E(I^r) \tag{1}$$

Previous research [13] has demonstrated that learning residuals is more feasible than learning the entire feature set directly. Moreover, the source image also possesses emotional attributes, which must be considered together with the reference emotion for effective

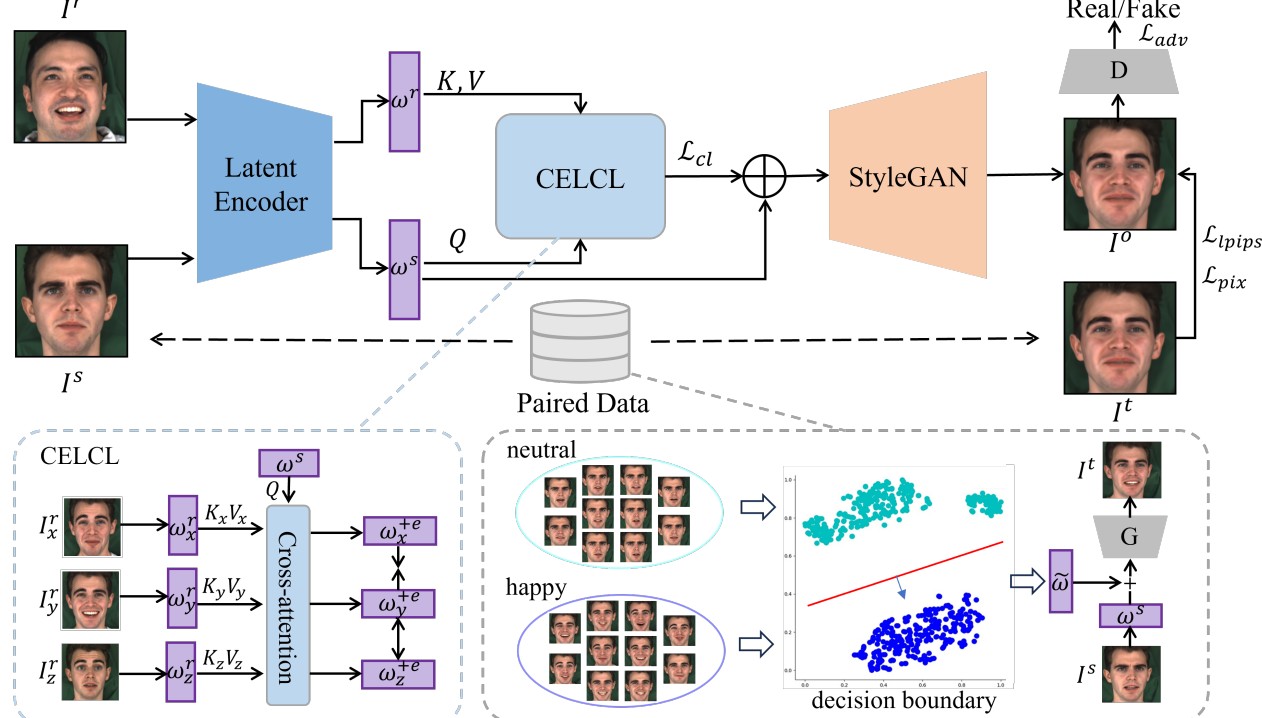

**Figure 1: An overall pipeline of the proposed Self-Supervised Emotion Representation Disentanglement framework. Given a source image and a reference image, the latent encoder predicts their latent codes. Following this, the CELCL module extracts emotion information and outputs emotion latent code. Finally, the StyleGAN takes the sum of source latent code and emotion latent code as input, and generates the output image. During training, we utilize synthetic paired data for additional supervision.**

manipulation. Therefore, we employ a multi-head cross-attention (MHCA) mechanism [33] to predict the emotional residual. It takes $\omega^s$ as query $\mathbf{Q}$, $\omega^r$ as key and value $\mathbf{K}, \mathbf{V}$ via

$$\begin{aligned} \mathbf{Q} &= \omega^s \mathbf{w}_q + \mathbf{b}_q \\ \mathbf{K} &= \omega^r \mathbf{w}_k + \mathbf{b}_k \\ \mathbf{V} &= \omega^r \mathbf{w}_v + \mathbf{b}_v \end{aligned} \qquad (2)$$

Then, we use a MHCA layer followed by several feed forward network (FFN) layers to fuse them

$$\omega^e = \text{FFN}\left(\text{MHCA}\left(\mathbf{Q}, \mathbf{K}, \mathbf{V}\right)\right) \qquad (3)$$

Since the distribution of the latent code after MHCA diverges from the original latent space of StyleGAN, additional nonlinear layers is essential for improved model fitting. Therefore, we adopt two transformer layers $\mathcal{T}^2$ [33] to refine the emotion latent code

$$\omega^{+e} = \mathcal{T}^2(\omega^e) \qquad (4)$$

Finally, the emotion latent code $\omega^{+e}$ is added to the source latent code $\omega^s$ and then fed into StyleGAN to generate an image that matches the reference emotion.

During training, it is expected that the emotion latent codes corresponding to the same emotion should be similar, whereas codes associated with different emotions should be dissimilar. Specifically, we define the emotion set with seven emotions followed by [22].

Given the emotion latent code $\omega^{+e}_{i,k}$ of $i$-th image with emotion $\epsilon_k, k \in [1,7]$, we calculate the contrastive loss as follows

$$\ell_i = -\log \frac{\exp\left(\omega^{+e}_{i,k} \cdot \omega^{+e}_{*,k}/\tau\right)}{\exp\left(\omega^{+e}_{i,k} \cdot \omega^{+e}_{*,k}/\tau\right) + \sum_{k'=1,k'\neq k}^{7} \exp\left(\omega^{+e}_{i,k} \cdot \omega^{+e}_{*,k'}/\tau\right)} \qquad (5)$$

where $\tau$ is a temperature coefficient that is set 0.1 in our experiments. The $\omega^{+e}_{*,k}$ and $\omega^{+e}_{*,k'}$ are cached emotion latent codes. To optimize memory usage, we cache emotion latent codes from previous iterations. When calculating the contrastive loss, we randomly select cached codes associated with specific emotions, categorizing them as either positive or negative samples. We calculate the summation over all images to obtain the final objective function, formulated as

$$\mathcal{L}_{cl} = \sum_i \ell_i \qquad (6)$$

## 3.2 Paired Data Construction

While the CELCL module is effective in extracting emotion information, the generator still depends on pixel-level supervision to produce the intended outcomes. Moreover, the absence of time-aligned paired data for the SPFEM task presents challenges in providing such detailed supervision. According to the previous researches [3, 16, 38], a pretrained StyleGAN showcases exceptional

capabilities in editing and controllable generation. By manipulating the latent code towards a specific direction within the latent space, it is capable to produce high-quality images with specific attributes. Therefore, we propose a paired data construction module that utilizes a pretrained StyleGAN to synthesize paired data.

Our approach begins with the training of a latent encoder designed for StyleGAN inversion, enabling the mapping of images to their corresponding latent codes. The structure of the latent encoder is consistent with $E$ in Section 3.1. Subsequently, we identify discriminative boundaries among the latent codes associated with faces expressing varying emotions, which serve as the emotion editing direction. It's crucial that the emotion editing direction is predominantly influenced by the emotion, rather than the mouth shape, since latent codes representing the same emotion but different mouth shapes should be categorized identically. Once the emotion editing direction is established, we edit the latent codes to generate paired data via StyleGAN.

More concretely, given an input image $I^s$, we derive latent code $\omega^s$ via Equation 1. Then we feed $\omega^s$ into StyleGAN to reconstruct the input image

$$I^{s'} = G(\omega^s) \tag{7}$$

where $G$ is the pretrained StyleGAN with frozen weights. The reconstructed image $I^{s'}$ is supposed to be consistent with $I^s$. The reconstruction loss is defined as follows

$$\mathcal{L}_{rec} = \left\| I^{s'} - I^s \right\|^2 + \alpha \left\| \phi_p(I^{s'}) - \phi_p(I^s) \right\|^2 \tag{8}$$

where $\phi_p$ denotes VGG19 [28] for calculating perception loss, and $\alpha$ is a balance factor that is set to 0.8. We also utilize generative adversarial loss to improve the realism of reconstruction

$$\mathcal{L}_{adv} = \min_G \max_D \mathbb{E}\left[ \log(D(I^s)) \right] + \mathbb{E}\left[ \log(1 - D(I^{s'})) \right] \tag{9}$$

Moreover, we introduce emotion contrastive loss to the latent code $\omega^s$ as defined in Equation 6. This objective function amplifies the distinction between latent codes associated with varying emotions, thereby enhancing their discriminability. The total loss for StyleGAN inversion can be defined as the sum of these losses

$$\mathcal{L}_{inv} = \mathcal{L}_{rec} + \lambda_1 \mathcal{L}_{adv} + \lambda_2 \mathcal{L}_{cl} \tag{10}$$

where $\lambda_1$ and $\lambda_2$ are balance factors that are both set to 0.1.

After training the latent encoder, it predicts latent codes for all the training images. Subsequently, we take the latent codes associated with a neutral emotion as the source and the latent codes associated with other emotions as references. Then we sequentially identify the boundaries between the neutral emotion and the other six emotions. Taking the happy emotion as an example, we label the latent codes corresponding to a neutral emotion as 0 and those corresponding to a happy emotion as 1. We then apply a Support Vector Machine (SVM) [5] to classify these latent codes and output a decision boundary. This decision boundary is subsequently normalized to define the direction for editing emotions. A brief overview of this process is illustrated in the bottom right of Figure 1. The formula can be defined as follows

$$\tilde{\omega} = \mathcal{N}(\phi_{svm}(\Omega, \mathbf{Y})) \tag{11}$$

where $\Omega$ represents the union of latent codes associated with the two emotions, while $\mathbf{Y}$ denotes the set of labels corresponding

to these latent codes. The function $\phi_{svm}$ signifies the application of a SVM to fit the input data and corresponding labels, and $\mathcal{N}$ represents the L2 normalization function. In this way, we obtain an editing direction $\tilde{\omega}$ that transitions from a neutral emotion to a happy emotion. Finally, we use the editing direction to produce pairs of neutral and happy images

$$I^t = G(\omega^s + \beta\tilde{\omega}) \tag{12}$$

where $I^t$ is the synthesized image expressing happy emotion and matching the mouth shape of $I^s$. $\beta$ is the intensity coefficient and is set to 15. Such process is repeated until paired data is constructed for each emotion.

We realize that this method also allows us to manipulate facial expressions in SPFEM task. However, the SPFEM task requires altering the emotion in a source image to match that of a reference image, while the way of editing latent code depends on a predetermined editing direction. The SPFEM approach offers greater flexibility in practical applications. Therefore, the emotion disentanglement in our framework is necessary for SPFEM and we can utilize this method to supervise our framework.

## 3.3 Hybrid Training Strategy

After establishing paired data, we leverage it to train our framework for SPFEM. Given a source image $I^s$, a reference image $I^{r_1}$ from the same speaker with $I^s$, another reference $I^{r_2}$ from a different speaker, and a target image $I^t$, our framework generates output images $I^{o_1}$ and $I^{o_2}$ that convey the emotion depicted in the reference images. Here, $I^s$ and $I^t$ are synthetic paired images that exhibit a consistent mouth shape yet differ in emotion. We utilize the target image to provide pixel-level supervision through pixel loss and perception loss [40]. The loss functions can be defined as follows

$$\mathcal{L}_{pix} = \left\| I^{o_1} - I^t \right\|^2 + \left\| I^{o_2} - I^t \right\|^2$$
$$\mathcal{L}_{lpips} = \left\| \phi_p(I^{o_1}) - \phi_p(I^t) \right\|^2 + \left\| \phi_p(I^{o_2}) - \phi_p(I^t) \right\|^2 \tag{13}$$

Particularly, $I^s$ is usually a synthetic image in the above training process, as the source images may represent any one of the seven emotions during training, and images depicting emotions other than neutral are synthetic. However, the synthetic images inevitably differ from real images, potentially undermining model performance. To further improve the image quality of SPFEM, we propose a hybrid training strategy that utilizing both sythetic paired images and real unpaired images for training. This method aims to mitigate overfitting to synthetic data, ensuring a more robust and effective model performance. Specifically, if the source image is synthetic, we utilize the target image for supervision; otherwise, we focus on optimizing the generative adversarial loss exclusively. The overall objective function can be defined as follows

$$\mathcal{L}_{all} = \mathbb{1}(I^s)\left[ \mathcal{L}_{pix} + \alpha\mathcal{L}_{lpips} \right] + \lambda_1 \mathcal{L}_{adv} + \lambda_2 \mathcal{L}_{cl} \tag{14}$$

where $\mathbb{1}(\cdot)$ is an indicator function that returns 0 for real images and 1 for synthetic images. $\alpha$, $\lambda_1$, $\lambda_2$ are balance coefficients that are set to 0.8, 0.1, 0.1, respectively.

| Datasets | Methods | Intra-ID | | | Cross-ID | | |
|---|---|---|---|---|---|---|---|
| | | FAD↓ | LSE-D↓ | CSIM↑ | FAD↓ | LSE-D↓ | CSIM↑ |
| MEAD | ICface | 6.795 | 10.083 | 0.775 | 9.540 | 11.238 | 0.688 |
| | EAT | - | - | - | 6.186 | 9.551 | 0.761 |
| | DSM | 2.152 | 9.531 | 0.806 | 4.460 | 9.917 | 0.778 |
| | NED | 2.108 | 9.454 | 0.831 | 4.448 | 9.906 | 0.773 |
| | Ours | **0.740** | **9.126** | **0.904** | **2.453** | **9.200** | **0.848** |
| RAVDESS | ICface | 8.443 | 8.480 | 0.755 | 9.424 | 11.539 | 0.677 |
| | EAT | - | - | - | 8.051 | 8.154 | 0.668 |
| | DSM | 2.354 | 7.653 | 0.871 | 4.258 | 8.209 | 0.756 |
| | NED | 3.057 | 7.562 | 0.825 | 5.412 | 8.034 | 0.760 |
| | Ours | **1.399** | **7.441** | **0.894** | **3.360** | **7.621** | **0.790** |

**Table 1: Comparision results of average FAD, LSE-D and CSIM of our framework and competing methods in the intra-identity and cross-identity settings on the MEAD and RAVDESS dataset.**

## 4 EXPERIMENTS

### 4.1 Experimental Settings

**Dataset.** MEAD [34] contains 60 speakers, where each speaker records 30 videos in each emotional state (i.e., neutral, happy, angry, surprised, fear, sad, and disgusted). We select 6 speakers (M003, M009, W029, M012, M030, and W015) that have 1,260 videos to train our framework. We randomly select 90% as the training set and the rest 10% as the test set. We also evaluate the performance of our framework on another widely-used RAVDESS dataset [20]. Specifically, we also select 6 speakers (actors 1-6) with 168 videos. Identically, 90% videos are randomly selected as the training set while the rest 10% are used as the test set.

**Implementation Details.** The preprocessing of the training and test data follows the method described by [22]. Subsequently, we resize the facial images to 320×320 pixels and apply a center crop to obtain 256×256 pixel images, ensuring tightly framed faces. Our framework is trained using the Ranger [36] optimizer over 50,000 iterations. We set the learning rate to 0.0001 with a batch size of 2. All models are trained on a single NVIDIA RTX 4090.

**Evaluation Protocol.** In this work, we evaluate using the following metrics. 1) Frechet Arcface Distance (FAD) [14] extract feature vectors of generated and real videos using a state-of-the-art face recognition network [6] and compute their difference to measure the realism of generated video. A small FAD value indicates better realism. This metric not only measures the clarity of images but also evaluates the naturalness of generated faces and the consistency of the expression style with real images. 2) Lip Sync Error-Distance (LSE-D) [23] computes the distance between the lip and audio representations via a pre-trained model [4], which can be used to evaluate the lip-audio preserving accuracy. 3) Cosine similarity (CSIM) [41] extracts their features using a state-of-the-art expression recognition network and computes their similarity to measure the emotion similarity between the generated video and target emotional video. A large CSIM value suggests higher similarity. We present the results of two settings, i.e., the intra-identity setting that the emotion reference and source video belong to the same speaker, and the cross-identity setting that belong to different speakers.

## 4.2 Comparison with State-of-the-art Methods

To evaluate the effectiveness of the proposed framework, we compare it with the following algorithms: 1) ICface (WACV 2020) [31] employs action units to depict facial expressions and to transition the source face to the target emotion. 2) DSM (ECCV 2022) [29] learns person-specific expression representation in the Valence-Arousal space and renders them as facial images. 3) NED (CVPR 2022) [22] combines the 3DMM parameters of the source identity and target emotion to achieve expression manipulation. 4) EAT (ICCV 2023) [10] employs a pretrained emotion-agnostic talking head transformer and integrates adaptation modules for emotional manipulation. Note that there are no intra-identity and cross-identity settings for EAT. This is because EAT employs predetermined emotional guidances to modify facial expressions, instead of extracting emotional cues from reference faces. To ensure a reasonably equitable comparison, we compare it against other methods under the cross-identity setting.

#### 4.2.1 Quantitative Comparisons.

The performance comparisons are illustrated in Table 1. ICface is a faical reenactment algorithm that adapts to SPFEM task. Although capable of altering expressions, it struggles to accurately preserve speech content. DSM, NED and EAT leverages the 3DMM or 3D keypoints to effectively preserve the original speech by controlling the mouth's movements. However, facial expressions encompass complexities that extend beyond what can be captured by 3D parameters/keypoints with limited number of dimensions. This limitation leads to sub-optimal image realism and emotion similarity. In contrast, our framework SSERD harnesses the StyleGAN latent space to comprehensively capture emotion information and to construct paired data for detailed supervision, resulting in higher image quality and more natural expressions while preserving the original shape of the mouth.

We first present the performance comparisons on the MEAD dataset. Due to the page limitation, we present the average results across seven emotions in Table 1. *Detailed results for each emotion are presented in the supplemental materials.* In intra-identity setting, our framework achieves superior performance in manipulating various expressions. Compared to the current SOTA method NED, which obtains average FAD, LSE-D, and CSIM of 2.108, 9.454, and 0.831, SSERD decreases the average FAD, LSE-D to 0.740, 9.126, and increases the CSIM to 0.904. These results indicate that our framework can generate more photorealistic images that well preserve the lip-audio synchronization while effectively manipulating target expressions. Cross-identity is a more general and practical setting, and our framework also shows obvious advantages on FAD, LSE-D, and CSIM as shown in Table 1. Compared to NED, our SSERD obtains a relative average FAD and LSE-D decrement of 44.9% and 7.1% and a relative average CSIM increment of 9.7%.

To demonstrate the robustness of the proposed framework, we also present the performance comparisons on the RAVDESS dataset. As shown in Table 1, SSERD also exhibits significant advantages for various expression manipulation in both settings. Compared to DSM, in intra-identity setting, SSERD decreases the average FAD, LSE-D by 0.703, 0.212, with a relative decrement of 29.9%, 2.8%. And it increases the average CSIM by 0.023, with a relative decrement of

| Source | Reference | ICface | DSM | NED | EAT | Ours |
|--------|-----------|--------|-----|-----|-----|------|

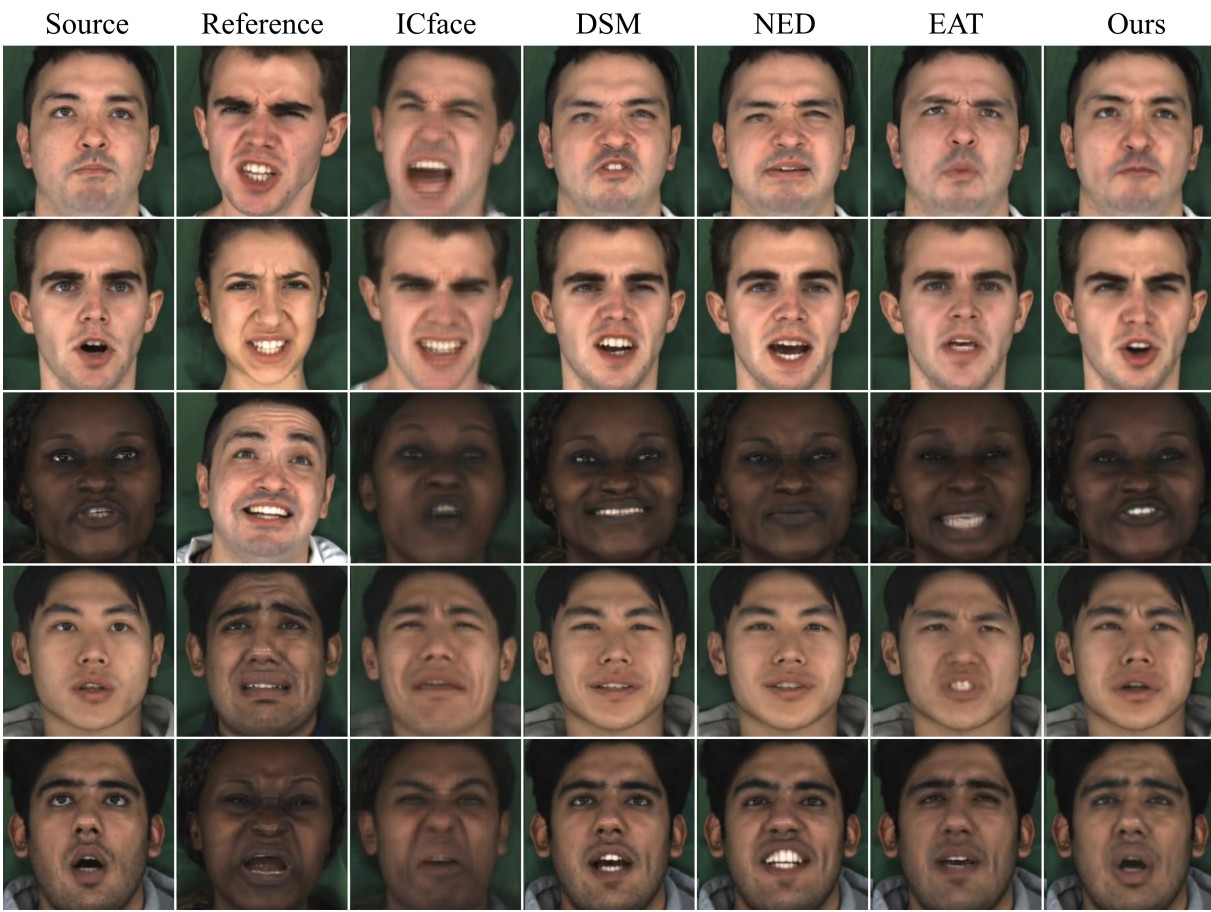

**Figure 2: Qualitative comparisons with state-of-the-art methods on the MEAD dataset. Our framework produces expressive talking faces featuring natural expressions and synchronized lip movements.**

2.6%. In cross-identification setting, SSERD decreases the average FAD, LSE-D by 0.898, 0.588 and increases CSIM by 0.034.

### 4.2.2 Qualitative Comparisons.

In this part, we present some visualization results of our framework and competing methods in Figure 2. Similar to the quantitative metrics, we analyze the qualitative comparisons from three aspects. 1) **Realism.** ICface produces blurred faces and suffers from identity confusion. DSM, NED and EAT yield faces that appear unnatural, particularly in the regions of the eyes and mouth, as evidenced in the first and third rows. In contrast, our framework generates faces with higher realism. 2) **Lip-audio preserving accuracy.** Due to the absence of time-aligned paired data, existing approaches struggle to accurately preserve mouth shapes in accordance with the source audio. This issue is evident in the second and fifth rows, where the mouth shapes in images generated by current methods markedly diverge from those in the source images. In contrast, by introducing paired data for supervision, our framework enhances the preservation of mouth shapes corresponding to the original audio. 3) **Emotion similarity.** Current methods utilize 3D parameters or 3D keypoints for emotion representation, which

are inadequate for capturing complex facial expressions. This limitation leads to significant discrepancies in facial expressions when compared to reference images, as evident in the third and fourth rows. In contrast, our framework harnesses the latent space of StyleGAN to achieve a more comprehensive representation of emotions, thus significantly improving the accuracy of generated expressions. *More visualization results on the MEAD and RAVDESS datasets are presented in the supplementary materials. For a more intuitive comparison, we have also included some video comparisons in the supplementary materials.*

### 4.3 User Study

We conduct web-based user studies to present the comparisons of our framework and the state-of-the-art methods. It consists of three parts corresponding to the above three metrics, i.e., realism, emotion similarity with the reference emotion, and mouth shape similarity with the source video. For each emotion, we select 10 videos from MEAD dataset and RAVDESS dataset, thus obtaining 70 videos. We find 20 participants to judge the three aspects of each video. As shown in Table 2, our framework surpasses other leading methods in all three metrics. Specifically, our framework achieves

| Methods | Realism | Emotion similarity | Mouth shape similarity |
|---------|---------|--------------------|------------------------|
| ICface | 1% | 1% | 1% |
| EAT | 14% | 19% | 11% |
| DSM | 26% | 20% | 25% |
| NED | 19% | 22% | 19% |
| Ours | **40%** | **36%** | **43%** |

Table 2: Realism, emotion similarity, and mouth shape similarity ratings of the user study.

ratings of 40% for realism, 36% for emotion similarity, and 43% for mouth shape similarity, outperforming NED by 21%, 14%, and 24%, respectively. *Detailed results for each emotion are available in the supplemental materials. All test videos utilized in the user study are included in the supplementary materials, facilitating verification and review.*

## 4.4 Ablation Study

The above comparisons with state-of-the-art methods well demonstrate the effectiveness of the proposed SSERD framework as a whole. In this part, we further delve into a detailed module to analyze their actual contributions. Here, we conduct the experiments on MEAD dataset.

### 4.4.1 Analysis of Contrastive Learning.

| Settings | Methods | FAD↓ | LSE-D↓ | CSIM↑ |
|----------|---------|------|--------|-------|
| Intra-ID | w/o $\mathcal{L}_{cl}$ | 0.923 | 9.134 | 0.891 |
|          | w/ $\mathcal{L}_{cl}$ | **0.740** | **9.126** | **0.904** |
| Cross-ID | w/o $\mathcal{L}_{cl}$ | 2.601 | 9.227 | 0.842 |
|          | w/ $\mathcal{L}_{cl}$ | **2.453** | **9.200** | **0.848** |

Table 3: Comparision results of average FAD, CSIM, and LSE-D of our framework with and without contrastive learning.

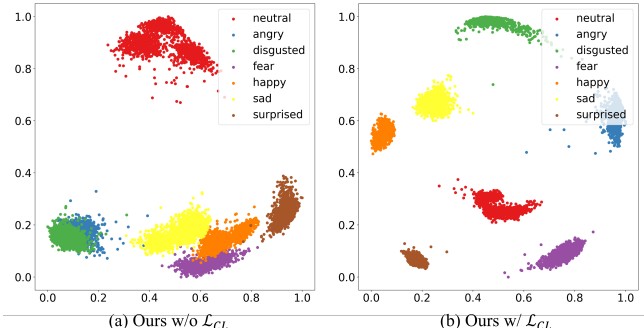

(a) Ours w/o $\mathcal{L}_{CL}$          (b) Ours w/ $\mathcal{L}_{CL}$

Figure 3: PCA visualization of emotion latent codes with and without contrastive learning.

Contrastive learning is employed to disantengle emotion-related information from reference faces and minimize irrelevant details. In our study, we removed this module to assess its impact. According to Table 3, incorporating contrastive learning significantly enhances the expression similarity and the realism of the generated images. For a clearer comparison, we visualize emotion latent codes using the PCA algorithm. Figure 3 reveals that, without contrastive learning, the emotion latent codes for anger (blue) and disgust (green) tend to merge. Additionally, the codes representing sadness (yellow), happiness (orange), and fear (purple) cluster closely together. In contrast, with contrastive learning, there's a marked improvement in the separation of latent codes for each distinct emotion, underscoring its effectiveness. Further qualitative analysis, as shown in Figure 4, indicates that contrastive learning results in more photorealistic facial expressions. Consequently, it not only improves emotion similarity but also enhances image realism by aligning the distribution of facial expressions more closely with real images.

| Source | Reference | w/o $\mathcal{L}_{CL}$ | Ours |
|--------|-----------|------------------------|------|

Figure 4: Qualitative comparisons of our framework with and without contrastive learning. Introducing contrastive learning results in more photorealistic facial expressions.

### 4.4.2 Analysis of paired data construction module.

| Settings | Paired Data | FAD↓ | LSE-D↓ | CSIM↑ |
|----------|-------------|------|--------|-------|
| Intra-ID | MEAD | **0.732** | 9.254 | **0.904** |
|          | NED | 1.463 | 9.160 | 0.874 |
|          | Ours | 0.740 | **9.126** | **0.904** |
| Cross-ID | MEAD | **2.431** | 9.342 | 0.849 |
|          | NED | 5.035 | 9.346 | 0.732 |
|          | Ours | 2.453 | **9.200** | 0.848 |

Table 4: Comparision results of average FAD, CSIM, and LSE-D of our framework with different paired data.

In this work, we construct paired data by editing latent codes for different emotions using a pretrained StyleGAN. In fact, MEAD

dataset includes some paired data. However, these data are not strictly aligned in time, complicating the implementation of frame-by-frame supervision. Here, we align the paired data in MEAD dataset by aligning audios via dynamic time warping [1] algorithm, and use the aligned data to train our framework for comparison. As shown in Table 4, using paired data of MEAD dataset results in diminished lip synchronization. This discrepancy arises because the paired data lacks precise alignment, and the mouth shapes corresponding to different emotions are sometimes inconsistent. Conversely, our method successfully produces well-aligned pairs, thereby more accurately preserving the original speech. Further-more, although using recorded paired data in MEAD dataset achieves a lower FAD, it increases the cost of video recording for practical application.

To further demonstrate the effectiveness of our proposed paired data construction module, we also use pretrained NED models to generate paired data for training. As shown in Table 4, utilizing paired data derived from NED leads to enhanced lip synchronization in Intra-ID setting when compared to using paired data from MEAD, albeit with a trade-off in terms of image realism and emotion similarity. Notably, our approach excels the NED in terms of FAD, LSE-D, and CSIM across both settings.

*4.4.3 Analysis of Hybrid Training Strategy.*

| Settings | Training Data | FAD↓ | LSE-D↓ | CSIM↑ |
|---|---|---|---|---|
| Intra-ID | Unpaired data | 3.052 | 9.419 | 0.877 |
| | Paired data | 0.835 | 9.155 | 0.900 |
| | Mixed data | **0.740** | **9.126** | **0.904** |
| Cross-ID | Unpaired data | 4.314 | 9.567 | 0.826 |
| | Paired data | 2.524 | 9.210 | 0.845 |
| | Mixed data | **2.453** | **9.200** | **0.848** |

**Table 5: Comparision results of average FAD, CSIM, and LSE-D of our framework with different training strategy. Using mixed data achieves better image realism.**

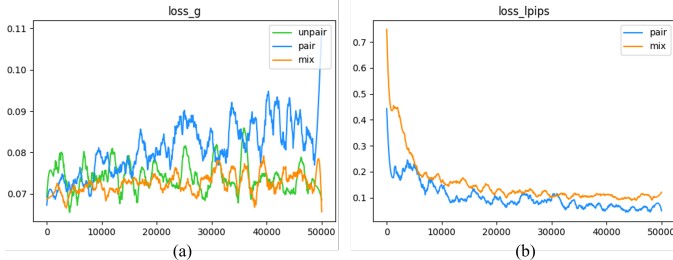

**Figure 5: Plot of loss curves. (a) Generative loss. (b) Perception loss.**

We conduct experiments using different training strategies to demonstrate the effectiveness of the proposed hybrid training strategy. As shown in Table 5, utilizing only unpaired data for training leads to sub-optimal performance, attributable to the

inherent unpredictability of GANs. Sole reliance on generative adversarial losses proves inadequate for producing the desired outcomes, highlighting the indispensability of detailed paired supervision for SPFEM. In contrast, the use of synthetic paired data markedly enhances performance across all evaluated metrics. Nonetheless, synthetic paired images inherently diverge from their real counterparts. As illustrated in Figure 5.(a), the generation loss associated with synthetic paired data (blue) increases during training, suggesting an increasing ease for the discriminator in identifying fake images. Conversely, the generation loss incurred from combining synthetic paired data with real unpaired data (orange) is close to that of employing solely real unpaired data (green), signifying a closer alignment of the model's outputs with real images. The results in Table 5 also show that using mixed data for training achieves better image realism. Moreover, we can further analyze this from the perception loss. As shown in Figure 5.(b), although using solely paired data yields better perception loss convergence, the quality of images generated by SPFEM model is heavily relied on the quality of synthetic paired data. In the contrary, introducing real data into the training process mitigates the overfitting to synthetic data, thereby improving the overall image realism.

## 5 LIMITATIONS

While our framework can generate photorealistic facial animations, it still faces some limitations, especially the constraint imposed by the limited set of emotions. In the current implementation, our approach relies on predetermined editing directions for seven specific emotions to synthesize paired data. This methodology enables the framework to effectively modify facial expressions to reflect these particular emotions; however, it does not support the creation of expressions outside this predefined set. This limitation restricts the versatility of our framework, confining the scope of expressiveness to a fixed vocabulary of emotional states. To address this constraint, future work could explore more flexible methods that encompass a wider and more varied range of emotions. For instance, we could leverage multi-modal large model techniques like CLIP [25] to depict an unlimited range of emotions through text descriptions.

## 6 CONCLUSION

This work presents a Self-Supervised Emotion Representation Disentanglement (SSERD) framework that focuses on disentangling emotion representation for accurate emotional cue transfer and develops a paired data construction module for effective supervision. The SSERD framework employs a contrastive learning module inte-grated with the latent space of StyleGAN for emotion representation and a cross-attention mechanism for emotion transference. This framework overcomes data pairing challenges by generating paired training data via a pretrained StyleGAN. It is further combined with a hybrid training strategy that leverages both synthetic and real data to enhance image realism. Extensive experiments conducted on various benchmarks have demonstrated the effectiveness of the proposed SSERD framework.

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
