# OpenReview forum: "Self-Supervised Emotion Representation Disentanglement for Speech-Preserving Facial Expression Manipulation"
_acmmm.org/ACMMM/2024/Conference — MM2024 Oral_

### Official Review · Reviewer_f5hu · 2024-05-21

**Rating:** 3
**Confidence:** 2

**Summary:**

In this paper, the authors propose a self-supervised emotion representation disentanglement (SSERD) method for speech-preserving facial expression manipulation. Specifically, the authors design an emotion latent learning module based on StyleGAN. Then, the authors employ a hybrid training strategy to improve the realism of generated images. Extensive experiments on two datasets are conducted to evaluate the effectiveness of the proposed method.

**Strengths:**

+ The motivation of designing the SSERD framework is interesting.
+ This paper is easy to read.
+ Extensive experiments on two benchmark datasets seem to be convincing.

**Limitations:**

- In Introduction, the authors point out that existing methods lead to the sub-optimal result along with two limitations. However, there are no relevant experimental results or evidence to support the above claims.
- In this paper, the authors consider paired facial expression data under the same identity. However, the collection of these data in real-world setting is difficult. I wonder to know the proposed method would work well using unpaired data.
- Some ablation studies are missing, for example, the effect of the balance factors in Eq. (10).
- Compared to Diffusion Model, what is the advantage of the proposed method? Also, what are its inspirations for this study?
- As discussed in the limitations section, the authors claim that they will leverage CLIP to describe emotions using different text prompts. As far as I know, the performance of CLIP on emotion recognition task is limited, such as [r1].
[r1] CLIPER: A Unified Vision-Language Framework for In-the-Wild Facial Expression Recognition.
- English proofreading is required. The usage of some sentences and words are not proper.

**Suitability:**

2

---

### Official Review · Reviewer_Jkrd · 2024-05-24

**Rating:** 5
**Confidence:** 3

**Summary:**

The paper focused on the task of Speech-Preserving Facial Expression manipulation (SPFEM) and proposed a new framework Self-Supervised Emotion Representation Disentanglement (SSERD). First, the authors proposed a contrastive emotion latent code learning in the latent space of StyleGAN. Second,  to mitigate the absence of strictly paired data, the authors leveraged a pretrained StyleGAN to generate paired data under the guidance of expression editing. Third, a hybrid training with real and synthetic data was deployed to avoid overfitting to synthetic data. Extensive experiments quantitatively and qualitatively show the superiority of the proposed method compared to existing ones.

**Strengths:**

1. The paper is well-motivated, clearly presented, easy-to-follow.
2. The proposed framework is interesting and novel and is shown effective by the experiments
3. I appreciate the qualitative and quantitative ablation study to better highlight the effectiveness of the proposed components.

**Limitations:**

1. For the emotion edition direction, I am wondering if the authors tried to vary the value of $\beta$ in eq.12 to see the impact on emotion editing. As ω represents the normalized direction, what if $\beta$ is very large?

2. I am wondering how the proposed method would do when it is fed with out-of-distribution samples. Since the latent encoder and the emotion editing direction reply to the distribution of the training dataset, would the model generalize to a face or expression that is not in the training distribution?

3. Authors might refer to traditional methods to edit face images while preserving lip shape, such as [1,2]. That seems related.

[1] Robust Face Frontalization For Visual Speech Recognition
[2] Expression-preserving face frontalization improves visually assisted speech processing

**Suitability:**

3

---

### Official Review · Reviewer_6wqc · 2024-05-26

**Rating:** 6
**Confidence:** 3

**Summary:**

This paper addresses the problem of speech-preserving facial expression manipulation in videos. The proposed approach incorporated a contrastive emotion latent code learning module for efficient representation of emotions and a paired data construction module for automatic synthesis of paired data. The paper reports significant improvements in this task using the MEAD and RAVDESS datasets.

**Strengths:**

- The paper proposes a new approach which leads to significant improvements
- The paper provides a thorough technical description of the proposed methodology
- The paper provides extensive evaluation and discussion of the obtained results

**Limitations:**

Apart from the limitations mentioned in the paper (limited set of emotions), I don't see any major weaknesses in the work. The paper is very thorough ad well written.

**Suitability:**

3

---

### Meta-Review · Area_Chair_DARv · 2024-07-01

**Recommendation:** Accept (Oral)
**Confidence:** 4

**Metareview:**

This meta-review summarizes the evaluations from reviewers 6wqc, Jkrd, and f5hu for submission 2226, titled "Self-Supervised Emotion Representation Disentanglement for Speech-Preserving Facial Expression Manipulation." Overall, the reviewers agree that the paper presents a novel and effective approach (SSERD) for the task and recommend accepting it.

**Strengths:**

* The paper proposes a new framework (SSERD) with significant improvements in speech-preserving facial expression manipulation. (all reviewers)
* The paper is well-written, clear, and easy to follow. (Jkrd, f5hu)
* The methodology is thoroughly described and the evaluation is extensive. (6wqc)
* The ablation study effectively highlights the contribution of each component. (Jkrd)

**Weaknesses:**

* The limitations section could be strengthened. While some reviewers found no major weaknesses (6wqc), others raised concerns about:
    * Lack of comparison with existing methods in the introduction (f5hu).
    * Generalizability to out-of-distribution data (Jkrd).
    * The use of CLIP for emotion recognition, considering its limitations (f5hu).
* Minor points include the need for English proofreading (f5hu) and exploring the impact of specific hyperparameters (Jkrd).

All reviewers agreed that the paper is relevant to the conference due to its contribution to multimedia processing.

Based on the reviews, I recommend accepting submission 2226 for presentation at ACM Multimedia 2024.